# Knowledge, attitude, and practice regarding dengue among non-health undergraduate students of Nepal

**Sheetal Bhandari**[1], **Manish Rajbanshi**[1]*, **Nabin Adhikari**[1], **Richa Aryal**[2],
**Kshitij Kunwar**[1], **Rajan Paudel**[1,3]

**1** Central Department of Public Health, Institute of Medicine, Tribhuvan University, Kathmandu, Nepal,
**2** Department of Public Health, Om Health Campus, Purbanchal University, Kathmandu, Nepal, **3** Unit of
Health Sciences, Faculty of Social Sciences, Tampere University, Tampere, Finland

* manishrajbanshi717@gmail.com

**Data Availability Statement:** The data is publicly available on data repository. URL: https://doi.org/10.6084/m9.figshare.25796230.v1.

## Abstract

Dengue poses a significant public health concern worldwide. It is identified as a recent emerging infectious disease in Nepal. Understanding the situation and dynamics between knowledge, attitudes, and practices (KAP) related to dengue among students is crucial for effective prevention and control strategies. This study aimed to assess the KAP and their associated factors of dengue among non-health undergraduate students of Nepal to identify gaps and suggest appropriate interventions. A web-based cross-sectional study was conducted among 429 non-health undergraduate students at eleven Nepalese Universities, with 80% of participants from the four most prominent ones in the country. Self-administered online forms were administered via Google Forms platform predominantly through social media for data collection. Data was cleaned and then exported to IBM SPSS Statistics 20.0 for analysis. Demographic characteristics of respondents were described using descriptive statistics. Multivariate logistic regression was conducted to determine the association between individual characteristics and KAP. Pearson's correlation coefficient was used to determine the association between knowledge-attitude, attitude-practice, and knowledge-practice. Statistical significance was determined at the P-value < 0.05. Around half of the participants were female (50.3%). The majority of participants were between 22 to 37 years, unmarried, and belonged to the Brahmin/Chhetri ethnic group. This study demonstrated a significant gap in KAP. Only 15.2% of participants had good knowledge while 25.9% and 68.3% of participants exhibited good attitudes and practices respectively. Marital status (AOR = 3.32, CI: 1.32–8.34), third-year educational level (AOR = 3.59, CI:1.34–9.57), and fourth-year educational level (AOR = 4.93, CI:1.88–12.94) were significantly associated with knowledge regarding dengue. Age (AOR = 1.73, CI: 1.10–3.01) was significantly associated with preventive practice regarding dengue. None of the demographic or socio-economic characteristics of respondents was associated with attitude on dengue. The knowledge-attitude ($r_{ka}$ = 0.01), knowledge-practice ($r_{kp}$ = 0.22), and attitude-practice ($r_{ap}$ = 0.01) were positively correlated in this study.

**Funding:** The author(s) received no specific funding for this work.

**Competing interests:** The authors have declared that no competing interests exist.

## Author summary

- Dengue has been identified as one of Nepal's youngest emerging infectious diseases. Nepal had its first dengue outbreak in 2006 followed by major cyclic outbreaks in 2010, 2013, 2016, 2019, and 2022.

- This study particularly tried to capture the situation of the knowledge, attitude, and practice of non-health undergraduate students and to possibly help identify the gaps and to support interventions planning to improving the endemic condition of dengue.

- This study found that 15.2% of participants had good knowledge, 25.9% and 68.3% of the participants had good attitudes and preventive practices regarding dengue respectively.

- To bridge the gap between knowledge and practice on dengue and promote KAP on dengue, television, and social media can be effective tools as these are major sources of information about dengue.

- Tailored interventions addressing demographic-specific factors are imperative for enhancing KAP on dengue prevention and control.

## Introduction

Dengue is considered one of the predominant arboviral infections caused by serotypes (DENV 1–4) and is transmitted through bites of infected *Aedes aegypti* [1]. In the last 50 years, the incidence of dengue has increased 30 times with increasing geographic expansion and has become a major public health concern globally [2]. Around 70% of dengue cases were reported in the Asian Region [3]. Dengue is endemic in more than 100 countries, mostly among the tropical and subtropical populated areas. It is being reported predominantly in the South-East Asia Region (SEAR), the Americas and the Western Pacific region [4].

Dengue has been identified as a recent emerging infectious disease in Nepal. A substantial increase in dengue incidence has been challenging to the health system of Nepal for the prevention and control of dengue [5]. Nepal had its first dengue outbreak in 2006 followed by the major outbreak in 2010, 2013, and 2016 [6]. Tracking cyclic outbreaks, 18,000 cases were reported in 68 districts in 2019 [7]. In 2022, there had been 54,784 reported dengue cases and 88 deaths which were almost three times higher compared to 2019 marking the largest number ever recorded in the country. As of July 15, 2023, 2,930 dengue cases and 8 deaths had been identified from 68 districts [8]. The ongoing response of the Government of Nepal (GoN) towards dengue including vector control needs to be improved [9,10].

Previous studies among non-health students showed that their knowledge of dengue was poor [11]. University students represent an engaged demographic group to gauge knowledge, perceptions, and practices against vector-borne diseases like dengue. Additionally, University students can be a good source for community preparedness that will help to share knowledge with families and establish sound dengue fever preventive and control practices for the community as a whole. We view undergraduate non-health students as a potential target population to tackle the growing dengue crisis, environmental control, and community engagement and for improving prevention and control.

Previous studies have focused on community members or health-related students, the latter likely to have more knowledge than the general public. This study particularly tried to capture the situation of the knowledge, attitude, and practice of non-health undergraduate students and to possibly help identify the gaps and to support interventions planning to improving the endemic condition of dengue. The findings of the study may be the point of reference to bridge the gap to design interventional strategies in collaboration with educational institutions, communities, local health authorities, and policymakers to establish effective vector control programs and to prepare for dengue outbreaks in the future.

## Methods

### Ethics statement

The approval for the study was obtained from the Institutional Review Committee (IRC) of the Institute of Medicine (IOM), Tribhuvan University, Nepal (Reference no; 216 [6–11] E2 077/078). Study objectives were clearly explained on the online form to the study participants before data collection. Informed consent was obtained from participants on the first page of the online questionnaire.

### Study design and setting

A web-based cross-sectional study was employed across all the provinces of Nepal. Data collection was carried out between January to April 2020.

### Study population

The study population was non-health undergraduates, older than 18 years of age. Undergraduates from engineering, agriculture, management, forestry, veterinary, arts, and information technology backgrounds were classified as non-health students. It included participants from 11 national Universities in Nepal, and more than 80% of the undergraduate and postgraduate students are concentrated in the following four top Universities: Tribhuvan University (75.94%), Purbanchal University (6.23%), Pokhara University (6.91%) and Kathmandu University (4.15%).

### Sample size calculation

Convenience sampling was adopted to recruit participants for data collection. Sample size was calculated using the formula $N = Z^2pq/d^2$ [12]. Assuming 50.0% prevalence (p) from a previous study [13], 5% margin of error (d), 95% confidence interval (CI), and 10% non-response rate, the total number of participants for this study was determined as 424 in this study.

### Data collection

A self-administered online questionnaire in English was distributed via Facebook, Gmail, WhatsApp, and Viber for data collection. No personally identifiable information was collected from study participants or included in the study analysis. Each participant was allowed to complete the form only once, maintained by setting the 'limit to one response' feature.

### Tools and measures

The study tool was obtained from a similar study conducted by Dhimal et.al in Nepal [14]. The pretesting was done among around 10% of the sample size of the study (n = 45) via online form. Cronbach's Alpha was used to assess the reliability coefficient which is a measure of the

internal consistency of the questionnaire. The result showed that Cronbach's Alpha coefficients of KAP domains were 0.80, 0.72 and 0.74 respectively.

The KAP questionnaire (S1 Questionnaire) consists of two sections, the first section is related to the socio-economic and demographic characteristics of participants which includes: age (in completed years), sex (male/female), marital status (married/unmarried), ethnicity (Brahmin/Chhetri, Janajati, Madhesi, Dalit, Muslims, and others), education level (firstyear, second-year, third-year, and fourth-year), family monthly income (<NRs.40,000 and ≥NRs.40,000) and Province (Koshi Province, Madhesh Province, Bagmati Province, Gandaki Province, Lumbini Province, Karnali Province and Sudurpaschim Province).

The second section consisted of questions related to the knowledge, attitude and practice of dengue. Altogether there were a total of 48 questions in this section. It included 24 questions on knowledge regarding signs/symptoms, mode of transmission and preventive practices on dengue.

In the attitude section, 6 questions were on serious illness, risk and prevention of dengue, strategies to prevent dengue, the breeding place of mosquitoes, and active participation of the community in the prevention of dengue. A total of 19 questions were asked on the preventive practices towards dengue among participants.

Participant's KAP on dengue was categorized as "good" if they scored 80% or higher and "poor" otherwise, respectively. Evaluation of knowledge was done by assigning score 1 for correct response and score 0 for incorrect response. Participants were categorized into having good knowledge if they scored 19 or above and poor knowledge score below 19 in this study [14].

Participants were asked to rate their attitude on dengue using a five-point Likert scale. The scale ranged from "Strongly disagree" to "Strongly agree". For each question, participants were assigned scores of 1, 2, 3, 4, and 5 corresponding to the following responses: Strongly Disagree, Disagree, Not Sure, Agree, Strongly Agree. Based on the total score, participants were categorized as having poor (score <24) and good attitudes (score ≥ 24).

There were a total of 19 questions related to practice. Each participant was assigned a score of 1 for positive practice and 0 for negative practice. Based on the total score, participants were categorized as having poor practice (score < 16) and good practice (score ≥ 16).

## Data management and analysis

Data collected online was downloaded as a spreadsheet. All the collected information was systematically compiled, coded, checked, and edited before exporting to IBM SPSS Statistics 20.0 IBM for analysis. The respondents' socio-economic and demographic characteristics were described using frequencies, percentages, median and interquartile ranges. We employed multivariate logistic regression analysis to investigate the association between individual characteristics and participants' KAP on dengue. The model was constructed with individual characteristics as independent variables and KAP served as the dependent variable, dichotomized into 'good' and 'poor' categories. The significance of associations was determined using P-values. Adjusted Odds Ratios (AOR) with 95% Confidence Interval (CI) were calculated to quantify the strength and direction of these associations. Pearson's correlation coefficient was used to determine the relationship between knowledge-attitude, attitude-practice, and knowledge-practice. The statistical significance was set at P-value <0.05.

## Results

### Individual characteristics of the participants

Out of 590 participants invited to participate in our study, 429 responded, yielding a response rate of 72.2%. The study included a nearly equal distribution of females (50.3%) and males

(49.7%). Over half of the participants (54.1%) were between the age of 22 and 37 years. The majority of participants (71.8%) belonged to Brahmin/Chhetri ethnic group followed by Janajati (15.6%). Nearly all of the participants (94.2%) were unmarried. This study represented all four years of undergraduates with the highest participation from fourth-year students (38.2%), followed by first-year (28.9%), third-year (19.6%) and second-year students (13.3%) respectively.

In this study, the majority of the participants (63.2%) belonged to families having a monthly income of below 303.03 USD. Most participants were from Lumbini Province (41.5%) and Bagmati Province (24.7%) as these provinces have the greatest number of colleges compared to other. (Table 1)

### Knowledge on dengue among the participants

The majority of the participants recognized fever (90.0%), headache (67.1%) and rash (59.7%) as major signs/symptoms of dengue. However, only 13.5% incorrectly stated that all mosquitoes could transmit dengue. Around (67.0%) were aware that *Aedes* mosquitoes transmit dengue. This study found that 43.6% and 34.5% of the participants believed dengue could be transmitted by flies/ticks and from the infected person with dengue, respectively. Most participants (71.3%) correctly observed that dengue could be transmitted by blood transfusion but 46.6% incorrectly thought transmission could occur through contaminated water and food. The majority (80.7%) of the participants were aware of the way of reducing mosquito bites by using mosquito screens, bed nets and while 81.4% were aware of being able to do so using insecticide sprays. (Table 2)

### Factors associated with knowledge on dengue among participants

Table 3 presents the results of multivariate logistic regression analysis investigating the factors associated with knowledge on dengue among the study participants. Only 15.2% of the participants had good knowledge on dengue. Married participants exhibited higher knowledge levels compared to the unmarried participants, indicating a significant association (AOR: 3.32, 95% CI: 1.32–8.34). Participants in their third and fourth University year demonstrated significantly higher knowledge levels compared to those in their first year (AOR: 3.59, 95% CI: 1.34–9.57; AOR: 4.93, 95% CI: 1.88–12.94, respectively).

Additionally, no significant associations were found between sex, age, and monthly income level with knowledge of dengue among the participants.

### Attitude on dengue among the participants

In this study, more than half of the participants (55.5%) agreed that dengue is a serious disease. Furthermore, one-third (32.4%) agreed that they were at risk of contracting dengue. More than half of the participants (54.1%) believed that dengue could be prevented. Half of the participants (47.8%) strongly agreed that controlling the breeding place is a good strategy to prevent dengue. While 45% of the participants strongly agreed that dengue breeds in stagnant water of pots and bottles. (Table 4)

### Factors associated with attitude on dengue among the participants

In this study, almost three-fourths of the participants (74.1%) had a poor attitude on dengue. However, the attitude was not found to be associated with any individual characteristics of the study participants.

**Table 1. Individual characteristics of the participants.**

| Characteristics | Numbers (n) | Percentage (%) |
|---|---|---|
| **Sex** | | |
| Female | 216 | 50.3 |
| Male | 213 | 49.7 |
| **Age group (in years)**<br>**Median ± IQR = (22.0±3.0)** | | |
| 18–21 | 197 | 45.9 |
| 22–37 | 232 | 54.1 |
| **Ethnicity** | | |
| Brahmin/Chhetri | 308 | 71.8 |
| Janajati | 67 | 15.6 |
| Madhesi | 24 | 5.6 |
| Dalit | 12 | 2.8 |
| Others (Marwari, Bangali) | 18 | 4.2 |
| **Marital status** | | |
| Unmarried | 404 | 94.2 |
| Married | 25 | 5.8 |
| **Educational level** | | |
| First-year | 124 | 28.9 |
| Second-year | 57 | 13.3 |
| Third-year | 84 | 19.6 |
| Fourth-year | 164 | 38.2 |
| **Monthly income of the family (in NRs)** | | |
| Below or equal to NRs. 40000 ($\leq$USD 303.03) | 267 | 63.2 |
| Above NRs. 40000 ($>$USD 303.03) | 162 | 37.8 |
| **Province** | | |
| Koshi Province | 61 | 14.2 |
| Madhesh Province | 33 | 7.7 |
| Bagmati Province | 106 | 24.7 |
| Gandaki Province | 20 | 4.7 |
| Lumbini Province | 178 | 41.5 |
| Karnali Province | 18 | 4.2 |
| Sudurpaschim Province | 13 | 3.0 |

Demographics of participants in a knowledge, attitude, and practices (KAP) study on Dengue. This table summarizes the demographic characteristics of the participants who participated in the study (Table 1).**Sources of information on dengue**

The most common sources of information on dengue among participants were television (16.2%) followed by social media (15.6%).

## Practice on dengue among the participants

Table 5 shows the frequency of use of common dengue prevention practices including preventing human/mosquito contact (73.2%), use of insecticide spray (77.9%), use of window screens (84.4%), eliminating standing water around the house (85.8%), cutting down the bushes in the yard (83.9%), using mosquito coils/repellent cream (82.1%), garbage removal (92.1%), disposal of water containing containers (90.2%), and covering body with clothes (88.6%). Additionally, 14.0% believed that even doing nothing could reduce mosquitoes.

**Table 2. Knowledge on dengue among participants.**

| Statements | Correct response n (%) |
|---|---|
| **Knowledge on signs/symptoms** | |
| Fever | 386 (90.0) |
| Headache | 288 (67.1) |
| Rash | 256 (59.7) |
| Nausea/vomiting | 219 (51.0) |
| Joint pain | 213 (49.7) |
| Muscle pain | 190 (44.3) |
| Pain behind eyes | 172 (40.1) |
| Back pain | 90 (21.0) |
| Diarrhea | 86 (20.0) |
| Stomach pain | 61 (14.2) |
| **Knowledge on transmission** | |
| All mosquito transmits dengue fever | 58 (13.5) |
| Aedes mosquito transmit dengue fever | 286 (66.7) |
| Flies and ticks transmit dengue fever | 187 (43.6) |
| Ordinary person-to-person contact transmits dengue fever | 148 (34.5) |
| Transmitted through food and water | 200 (46.6) |
| Transmitted by blood transfusion | 306 (71.3) |
| Mosquito breed in stagnant water | 365 (85.1) |
| Mosquito screens and bet nets reduce mosquito | 346 (80.7) |
| Insecticide sprays reduce mosquito and prevent dengue | 349 (81.4) |
| Tightly covering water containers reduce mosquitoes | 334 (77.9) |
| Removal of standing water can prevent mosquito breeding | 378 (88.1) |
| Mosquito repellants prevent mosquito bite | 313 (73.0) |
| **Can you identify *Aedes* mosquito** | 167 (38.9) |
| **Dengue mosquito feeding time** | |
| Day time | 192 (44.8) |
| Night time | 237 (55.2) |

Knowledge on dengue among participants. This table summarizes participants' understanding of dengue signs/symptoms, transmission methods, and preventive measures (Table 2).

## Factors associated with practice on dengue among the participants

Table 6 displays the results of multivariate logistic regression analysis examining the factors associated with the practice of dengue prevention and control among the study participants. More than two-thirds of the participants (68.3%) had good dengue prevention and control practices in this study. Participants aged 22–37 years demonstrated a significantly higher likelihood of practicing good dengue prevention and control measures compared to those aged 18-21(AOR: 1.73, 95% CI: 1.10–3.01). Additionally, no significant associations were observed between sex, marital status, and different educational levels with dengue prevention and control practices among the participants.

## Correlation between knowledge, attitude and practice on dengue of the study participants

This study found a positive correlation coefficient between knowledge, attitude and practice on dengue ($r_{ka} = 0.01$, $r_{kp} = 0.22$, $r_{ap} = 0.01$). (Table 7)

**Table 3. Factors associated with knowledge on dengue among the participants.**

| Characteristics | Knowledge | | | Multivariate logistic regression | | |
| --- | --- | --- | --- | --- | --- | --- |
| | | | | Good knowledge | | |
| | Good n (%) | Poor n (%) | P-value | AOR | 95% CI | P-value |
| **Level of knowledge** | 65 (15.2) | 364 (84.8) | | | | |
| **Age** | | | | | | |
| 18–21 | 21 (10.7) | 176 (89.3) | 0.01 | 1.0 (ref.) | | 0.49 |
| 22–37 | 44 (19.0) | 188 (81.0) | | 0.77 | 0.37–1.61 | |
| **Sex** | | | | | | |
| Male | 34 (16.0) | 179 (84.0) | 0.6 | 1.0 (ref.) | | 0.50 |
| Female | 31 (14.4) | 185 (85.6) | | 0.82 | 0.47–1.44 | |
| **Marital status** | | | | | | |
| Unmarried | 56 (13.9) | 348 (86.1) | 0.005 | 1.0 (ref.) | | 0.01 |
| Married | 9 (36.0) | 16 (64.0) | | 3.32 | 1.32–8.34 | |
| **Education level** | | | | | | |
| First-year | 8 (6.5) | 116 (93.5) | 0.00 | 1.0 (ref.) | | 0.008 |
| Second-year | 5 (8.8) | 52 (91.2) | 0.5 | 1.53 | 0.47–4.96 | 0.47 |
| Third-year | 14 (16.7) | 70 (83.3) | 0.02 | 3.59 | 1.34–9.57 | 0.01 |
| Fourth-year | 38 (23.2) | 126 (76.8) | 0.00 | 4.93 | 1.88–12.94 | 0.001 |
| **Monthly income** | | | | | | |
| Below or equal to NRs. 40000 (≤USD 303.03) | 36 (13.5) | 231 (86.5) | 0.2 | 1.0 (ref.) | | 0.29 |
| Above NRs. 40000 (>USD 303.03) | 29 (17.9) | 133 (82.1) | | 1.35 | 0.77–2.36 | |

A summary of factors associated with knowledge on dengue among participants (Table 3). Good knowledge refers to a higher understanding of dengue compared to poor knowledge.

## Discussion

Undergraduate students represent a significant segment of the young population and can serve as key agents in spreading accurate information about dengue, thereby contributing to raising awareness and fostering behavioral changes within their communities [15].

This study showed that 15.2% of the students exhibited good knowledge which is quite comparable to a similar study in Nepal [14] but contrasts with the studies conducted in some other countries (Jamaica, Bangladesh and Malaysia) [11,16,17]. Although lower, students in Nepal had higher levels of knowledge compared to the general population which could be due

**Table 4. Attitude on dengue among the participants.**

| Statements | Strongly agree n (%) | Agree n (%) | Disagree n (%) | Strong disagree n (%) | Not sure n (%) |
| --- | --- | --- | --- | --- | --- |
| Dengue fever is a serious disease | 160 (37.3) | 238 (55.5) | 17 (4.0) | 2 (0.5) | 12 (2.8) |
| Risk of getting dengue | 33 (7.7) | 139 (32.4) | 82 (19.0) | 25 (5.8) | 150 (35.0) |
| Dengue fever can be prevented | 165 (38.5) | 232 (54.1) | 5 (1.2) | 0 (0) | 27 (63.0) |
| Good strategy to prevent dengue is controlling the breeding place | 205 (47.8) | 186 (43.4) | 13 (3.0) | 4 (0.9) | 21 (4.9) |
| Stagnant water in pots/bottles are breeding places for Aedes | 193 (45.0) | 186 (43.4) | 9 (2.1) | 3 (0.7) | 38 (8.9) |
| Communities are responsible for controlling the vector of dengue | 269 (62.7) | 136 (31.7) | 6 (1.4) | 2 (0.5) | 16 (3.7) |

A summary of participants' attitudes on dengue (Table 4). This table explores how participants view the seriousness of dengue, their perceived risk of getting it, and their beliefs about prevention strategies.

**Table 5. Practice on dengue among the participants.**

| Statements | Correct response n (%) |
|---|---|
| Use insecticide spray | 334 (77.9) |
| Prevent mosquito man contact | 314 (73.2) |
| Use professional pest control | 294 (68.5) |
| Use window screens | 362 (84.4) |
| Eliminate standing water around the house | 368 (85.8) |
| Cut down bushes in the yard | 360 (83.9) |
| Using mosquito eating fish | 216 (50.3) |
| Using mosquito's coils/mosquito repellent cream | 352 (82.1) |
| Removal of garbage/trash | 395 (92.1) |
| Disposing water holding containers such as tires, parts of automobiles, plastic bottles, crack pots etc. | 387 (90.2) |
| Use of fan to drive repelling mosquitoes | 269 (62.7) |
| Use of smoke to drive away mosquitoes | 206 (48.0) |
| Covering body with clothes | 380 (88.6) |
| Do nothing to reduce mosquitoes | 60 (14.0) |
| Eliminating mosquito breeding sites | 373 (86.9) |
| Covering water containers in house | 391 (91.1) |
| Government spray insecticide for controlling mosquitoes | 270 (62.9) |
| Turning containers upside down to avoid water collection | 309 (72.0) |
| **Frequently cleaning water-filled containers and ditches around the house** | |
| Always | 44 (10.3) |
| Often | 86 (20.0) |
| Sometimes | 290 (67.6) |
| Never | 9 (2.1) |

Practices on dengue prevention among participants (Table 5). This table summarizes the different methods participants use to prevent dengue mosquito breeding and bites.

to the students being the major group particularly involved within the internet and social media to explore health-related issues. Another reason could be that students are also connected to different groups and sectors where they can access and exchange health-related information from various sources. However, our study has different outcomes than reported with similar studies conducted in Indonesia, India, Nepal, and Malaysia [13,18–20]. Thus, comparatively good knowledge among the students reflects that they have an adequate awareness level that can be used to aware communities. Our findings also demonstrated that there is a significant gap in the KAP on dengue among undergraduate students, demonstrating an urgent need for improved public education and outreach about these issues in Nepal, a country which is experiencing a surge in cases and deaths due to dengue since 2022.

Most of the respondents were able to correctly identify typical symptoms of dengue. Among them, fever (90%) and headache (67.1%) were the most frequently mentioned symptoms and that is similar to the studies conducted in Jamaica, Bangladesh, Sri Lanka, India, and Nepal [11,16,18,19,21].

Half of the participants failed to mention other recognized symptoms of dengue such as pain behind the eyes and back pain. It could be because most participants were not experienced with the disease or had not witnessed a case from a close relative or a friend previously. Additionally, almost one-third (34.5%) of respondents also believed that ordinary person-to-

**Table 6. Factors associated with practice on dengue among the participants.**

| Characteristics | Practice | | | Multivariate logistic regression | | |
| --- | --- | --- | --- | --- | --- | --- |
| | | | | Good practice | | |
| | Good n (%) | Poor n (%) | P-value | AOR | (95% CI) | P-value |
| **Level of practice** | 293 (68.3) | 136 (31.7) | | | | |
| **Age** | | | | | | |
| 18–21 | 123 (62.4) | 74 (37.6) | 0.01 | 1.0 (ref.) | | 0.04 |
| 22–37 | 170 (73.3) | 62 (26.7) | | 1.73 | 1.10–3.01 | |
| **Sex** | | | | | | |
| Male | 149 (70) | 64 (30) | 0.46 | 1.0 (ref.) | | 0.77 |
| Female | 144 (66.7) | 72 (33.3) | | 0.94 | 0.61–1.43 | |
| **Marital Status** | | | | | | |
| Married | 15 (60) | 10 (40) | 0.35 | 1.0 (ref.) | | 0.20 |
| Unmarried | 278 (68.8) | 126 (31.2) | | 0.57 | 0.24–1.34 | |
| **Educational level** | | | | | | |
| First-year | 80 (64.5) | 44 (35.5) | 0.27 | 1.0 (ref.) | | 0.92 |
| Second-year | 36 (63.2) | 21 (36.8) | | 0.86 | 0.44–1.67 | 0.66 |
| Third-year | 61 (72.6) | 23 (27.4) | | 1.08 | 0.56–2.09 | 0.80 |
| Fourth-year | 116 (70.7) | 48 (29.3) | | 0.92 | 0.48–1.76 | 0.82 |
| **Monthly income** | | | | | | |
| Below or equal to NRs. 40000 (≤USD 303.03) | 187 (70.0) | 80 (30.0) | 0.3 | 1.0 (ref.) | | 0.33 |
| Above NRs. 40000 (>USD 303.03) | 106 (65.4) | 56 (51.4) | | 0.80 | 0.52–1.24 | |

Factors associated with practices on dengue prevention among participants (Table 6). This table summarizes the association between participant characteristics and their dengue prevention practices. 'Good practice' refers to participants who adopted more preventative measures.

person contact could also transmit dengue, possibly due to confusion with other communicable diseases such as COVID-19 and influenza.

Our study reported that most people were unaware of the transmission of dengue. Only 38% could correctly identify the vector (Aedes mosquito) responsible for causing dengue. This finding was supported by similar studies conducted in Sri Lanka [21] and Nepal [15].

This study demonstrated a positive correlation between knowledge, attitude, and practice. This finding is consistent with similar studies conducted in Nepal, Thailand and Jamaica [16,19,22]. This suggests an opportunity for proportionately improving all domains of KAP by working on one or more domain improvements. Another reason could be that the higher the health literacy, the more positive the attitude, and preventive practices.

**Table 7. Correlation between knowledge, attitude and practice on dengue.**

| | Knowledge | Attitude | Practice |
| --- | --- | --- | --- |
| Knowledge | 1 | 0.010 | 0.221 |
| Attitude | 0.010 | 1 | 0.013 |
| Practice | 0.221 | 0.013 | 1 |

Correlation between knowledge, attitude and practice on dengue (Table 7). This table shows the strength of the correlation between participants' knowledge of dengue, their attitude about dengue prevention, and the practices they take to prevent dengue.

Most of the participants reported that television, social media and teachers were predominant sources on accessing information regarding dengue which is similar to the findings reported by Jamaica, India and Malaysia [16,18,20].

This study found that knowledge was significantly associated with marital status and educational level, consistent with the findings from studies conducted in Malaysia, Nepal and Indonesia [13,17,19]. It might be due to the high health literacy level among adults. This is supported by a study done in Malaysia which revealed that educational intervention was effective in creating positive awareness about dengue [20] and was associated with preventive practices regarding dengue. This implies that younger participants were more likely to engage in dengue preventive practices.

This study demonstrated that married participants had a better level of knowledge than unmarried. It could be that married couples often share their daily experiences including health information, and if one partner has access to health-related information or works in the health sector, they may share their knowledge with family members.

This study highlighted that the current level of KAP on dengue demands effectively designed community-based interventions to improve prevention and control strategies regarding dengue.

### Limitations

Due to the COVID-19 pandemic, this study was conducted online, this limited to participate the students without internet access. This study relies on self-report measures; therefore, it is possible that some participants may not have responded accurately. This study was limited with respect to environmental factors like climatic conditions, temperature, precipitation, and rainfall which could influence the prevalence of dengue.

### Conclusion

Around 15.2% of non-health undergraduates in Nepal have good knowledge on dengue while 25.9% and 68.3% of them have good attitudes and good practices respectively. This suggests the gap in the knowledge, attitude, and practices (KAP) on dengue among undergraduate students. There were significant gaps in understanding certain aspects of the disease transmission and symptoms among non-health undergraduates. To control this recently emerging disease, community-based interventions, and school health education can be the medium to help spread awareness on dengue.

To bridge the gap between knowledge and practice on dengue and promote KAP on dengue, television and social media can be effective tools as these are major sources of information about dengue. Tailored interventions for different demographic groups such as the young and old, married and unmarried are necessary to improve knowledge and encourage dengue prevention practices across the entire population. There was a positive correlation between knowledge, attitude, and practice, indicating that enhancing knowledge could lead to more favorable attitudes and practices toward dengue prevention.

### Recommendation

The federal government should support schools and Universities in implementing health programs by providing guiding documents while also expanding its national dengue awareness campaign, targeting high-risk communities and planning tailored demographic interventions in collaboration with local governments.

Local governments can help promote KAP on dengue by implementing targeted educational and awareness campaigns, practical training on dengue prevention practices, tailoring

interventions to specific demographic groups, supporting school-based preventive programs, and engaging in community-led actions for dengue prevention and improving preventive behaviors. These measures can significantly reduce dengue morbidity and mortality, promoting public health and well-being.

Encouraging knowledge sharing within families, particularly between married couples could produce effective results on dengue prevention. Further studies on the epidemiology transitions of dengue, environmental factors and social factors need to be explored to better understand dengue prevention and management.

## Supporting information

**S1 Questionnaire. Knowledge, Attitude and Practices regarding Dengue among non-health undergraduate students of Nepal.**
(DOCX)

## Acknowledgments

We would like to acknowledge all the faculty of the Central Department of Public Health, Institute of Medicine, Tribhuvan University, Nepal for their guidance during the research project. Our appreciation goes to all individuals responding to the questionnaire for their participation.

## Author Contributions

**Conceptualization:** Sheetal Bhandari.

**Formal analysis:** Sheetal Bhandari, Manish Rajbanshi.

**Investigation:** Sheetal Bhandari.

**Methodology:** Sheetal Bhandari, Manish Rajbanshi, Richa Aryal, Kshitij Kunwar.

**Resources:** Sheetal Bhandari.

**Supervision:** Rajan Paudel.

**Writing – original draft:** Sheetal Bhandari.

**Writing – review & editing:** Manish Rajbanshi, Nabin Adhikari, Richa Aryal, Kshitij Kunwar, Rajan Paudel.

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
