## [Decision Letter · Decision Letter 0]

30 Oct 2023

Dear Mr. Rajbanshi,

Thank you very much for submitting your manuscript "Knowledge, attitude and practice regarding dengue among non-health undergraduate students of Nepal" for consideration at PLOS Neglected Tropical Diseases. As with all papers reviewed by the journal, your manuscript was reviewed by members of the editorial board and by several independent reviewers. In light of the reviews (below this email), we would like to invite the resubmission of a significantly-revised version that takes into account the reviewers' comments. 

This is an important subject, but the manuscript requires susbstantial revisions.

We cannot make any decision about publication until we have seen the revised manuscript and your response to the reviewers' comments.

We cannot make any decision about publication until we have seen the revised manuscript and your response to the reviewers' comments. Your revised manuscript is also likely to be sent to reviewers for further evaluation.

Sincerely,

Mei L. Trueba, PhD

Guest Editor

Audrey Lenhart

Section Editor

This is an important subject, but the manuscript requires susbstantial revisions.

We cannot make any decision about publication until we have seen the revised manuscript and your response to the reviewers' comments.

Reviewer's Responses to Questions

**Key Review Criteria Required for Acceptance?**

**Methods**

-Are the objectives of the study clearly articulated with a clear testable hypothesis stated?

-Is the study design appropriate to address the stated objectives?

-Is the population clearly described and appropriate for the hypothesis being tested?

-Is the sample size sufficient to ensure adequate power to address the hypothesis being tested?

-Were correct statistical analysis used to support conclusions?

-Are there concerns about ethical or regulatory requirements being met?

Reviewer #1: Sample size calculation: How the authors confirm the equal participation from the provinces or considered other factors for the selection of the samples to generalized the results. 

Discuss in briefly about the sample size calculation using the formula with non-response rate

Study population age group: is it 18-22 or up to 25?

Ethical clearance: Is the clearance from Institute of Medicine (IOM), Tribhuvan university is applicable for all University? What about NHRC clearance for this study.

Reviewer #2: -Are the objectives of the study clearly articulated with a clear testable hypothesis stated? Yes

-Is the study design appropriate to address the stated objectives? Yes

-Is the population clearly described and appropriate for the hypothesis being tested? Yes

-Is the sample size sufficient to ensure adequate power to address the hypothesis being tested? Yes

-Were correct statistical analysis used to support conclusions? I am not a statistician but in my understanding, analysis need to be conducted multivariate analyses

-Are there concerns about ethical or regulatory requirements being met? Yes

**Results**

-Does the analysis presented match the analysis plan?

-Are the results clearly and completely presented?

-Are the figures (Tables, Images) of sufficient quality for clarity?

Reviewer #1: Results: I suggest to highlights the important result findings of table rather than explaining too much below the particular table

- - P<00.5 significant value indicates by asterisk rather than bold in table

- Please construct the 2x2 table throughout the manuscript as far as possible. including table 5

There are too many irreverent tables. It is suggested to remove tables if there are no significant results including table 5. 

Fig: 1 It is suggested for further formatting

It is some how difficult to understand the knowledge in relation to marital status? Could you explain? 

Discussion needs rewriting to discuss finding in relation to previous publications and high lighting the major issues to be focused since the disease is first reported 2004. 

It is also suggested to have clear recommendation for the local and national government based on the finding.

Reviewer #2: -Does the analysis presented match the analysis plan? As mentioned, instead of univariate, multivariate analysis would be better

-Are the results clearly and completely presented? These will change according to the analysis

-Are the figures (Tables, Images) of sufficient quality for clarity? Yes

**Conclusions**

-Are the conclusions supported by the data presented?

-Are the limitations of analysis clearly described?

-Do the authors discuss how these data can be helpful to advance our understanding of the topic under study?

-Is public health relevance addressed?

Reviewer #1: The conclusion was given, but not much was supported by the findings. There are some descriptions mentioned on the limitation that need further clarification. It has discussed the usefulness of the data. However, further recommendations for the system should be mentioned in more detail.

Reviewer #2: -Are the conclusions supported by the data presented? This may change after the multivariate analysis

-Are the limitations of analysis clearly described? Please see my comments

-Do the authors discuss how these data can be helpful to advance our understanding of the topic under study? Yes

-Is public health relevance addressed? Yes

**Editorial and Data Presentation Modifications?**

Reviewer #1: (No Response)

Reviewer #2: (No Response)

**Summary and General Comments**

Reviewer #1: Major comments: This MS was prepared based on a web-based interview with a non-health undergraduate student from four universities in Nepal. I wonder why the four universities selected the undergraduate non-health group was selected in this study. The other point is the language of the questioners: is it in English or Nepali? I could find if the author performed preformed pretesting and validation the questioners. The questions are so specific that they can only be understood by health workers, and these seem to be leading questions that have different outcomes. How have you defined non-health and health groups? Justification for adding value to this paper in relation to previous publications, including Dhimal et al. and others? There are lots of redundancies and duplications in the MS and the tables. The other important and pertinent issue is that there is not much emphasis on climatic factors, including temperature, precipitation, or rainfall.

Minor comments

• Write complete form of KAP in the initially in the abstract.

• Page 4, line 68 mentioned focused based awareness including street families. I am not aware of the such families in Nepal! Can author clarify. 

• Typographical error on page line 121 term strongly used twice? Needs appropriate correction.

• Line 149-150 error 94.2% married does not correlate with the Table 1. 

• The meaning is not understandable on Page 20 lane 216. It is suggested to rewrite. 

• It seems appropriate attention was not given preparing MS since several there are careless mistakes.

• Grammatical errors should be addressed properly.

Reviewer #2: The study is important as Dengue is considered endemic in Nepal and is associated with significant socioeconomic burden for the country. The study is well conducted but few issues needs to be considered before publication.

 As the sampling was convenience, how many students were approached for the survey? What was the response rate?

 The analysis consists of several univariate analyses using chi squared test and correlation. Multivariate analysis using logistic regression would be more appropriate to identify independent predictors of knowledge, attitude and practice.

 In the discussion section, it would be good to discuss the stigma associated with the disease regarding common misconception that the disease can spread directly from person to person.

 The limitation of convenience sampling should be discussed in the limitation section.

 The questionnaire needed to be provided in the supplementary file.

PLOS authors have the option to publish the peer review history of their article (what does this mean?). If published, this will include your full peer review and any attached files.

Reviewer #1: No

Reviewer #2: Yes: Professor Priya Paudya
---

## [Decision Letter · Decision Letter 1]

1 Apr 2024

Dear Mr. Rajbanshi,

Thank you very much for submitting your manuscript "Knowledge, attitude, and practice regarding dengue among non-health undergraduate students of Nepal" for consideration at PLOS Neglected Tropical Diseases. As with all papers reviewed by the journal, your manuscript was reviewed by members of the editorial board and by several independent reviewers. In light of the reviews (below this email), we would like to invite the resubmission of a significantly-revised version that takes into account the reviewers' comments. 

Section Editor Comments

I have taken over for Dr. Lenhart and taken the opportunity to provide another review of the revised manuscript. Below I try to provide a roadmap that will get this manuscript accepted for publication as all reviewer’s saw relevance and value in the publication. I have also taken the opportunity to edit the manuscript using the word version provided by the authors. All my suggestions are optional, and my assumption is that most grammatical errors were removed but some additional editing will improve understanding and the flow of the manuscript. Additionally, I include some comments to the review responses, and argue for inclusion of more of those responses in the body of the manuscript.

Previous Review Responses:

Reviewer #1 had a few questions associated with sample size and study design. (Sample size calculation: How did the authors confirm equal participation from the provinces or consider other factors for the selection of samples to generalize the results?, Study population age group: is it 18-22 or up to 25?) and Reviewer #2 asked how many students were approached.

Editor: To better address these concerns more details are needed in the study design and setting section. Principally, how did you approach students and who did you choose? I’m guessing you were able to email students inviting them to go to a link taking you to the online form. Alternatively, some kind of advertisement asking students to participate was distributed. It looks like students were recruited from the top four universities in Nepal. It would be easier to justify your approach, by simply stating that our aim was to understand Knowledge, Attitudes, and Practices of non-health oriented students and we selected these Universities because the had students from the widest cross section of provinces across Nepal and had access to the internet, etc. (not trying to put words in your mouth, but give us the reason you selected this population). This may represent the best and brightest and that would be something you would point out but seems like a reasonable starting point. Getting a perfectly representative sample is always very difficult so don’t claim that was your aim. 

• You need to better describe how you made contact with students and incorporate your response to Reviewer #2 about the response rate to the questionnaire.

• Report the range of age, from your response one still asks if the range is from 19-22 or 19-25. For example, in Table 1 instead of Less than 22 and 22 or above include the ranges, i.e. 19-21 versus 22-25.

• For provinces, perhaps you can included Bagmati and Lumbini (assuming they are closest to the named universities) and Other (you can list the as a footnote).

• When updating this sections, please give the reasons why you selected these 4 universities and what segment of the Nepalese population they are likely to represent.

• State in the methods that the form was administered in English.

• Add your response to Major comment from Reviewer #1. That is add a section to the methods on validation and pretesting of the study instrument.

• Include the definition of non-health students (also in the response to Major comments Reviewer #1) in this section.

• Ethical clearance: I am not as concerned about this, but all individuals who worked with identifiable information (PII) should have approval from each person’s institution or have those institutions defer to Tribhuvan University. Usually this is done through what is called a relying agreement or a letter indicating this was okay with each of the other institutions.

• In the Data management and analysis section, you should state that you carried out a multivariate analysis, but no significant differences were found and will not be discussed more, however this suggests that some of your univariate findings are due to confounding and should be interpreted with caution.

• In addition, to the inclusion of the questionnaire, you should indicate what you scored as a “correct” answer. In regard to this in Table 2, perhaps you could highlight statements that are incorrect in another color to distinguish with statements that are true. I will put the incorrect statements in red.

Other comments and observations:

• “youngest” emerging infection. Note sure what you mean by that. Do you a “recent”?

• Abstract – see some of my edits in yellow highlight or by comment. I have some concerns about Lines 40-42 and Lines 43-49.

• Introduction: see edits in yellow highlight.

• Methods: While you clearly describe how knowledge was scored (80%), it is not clear how you scored Attitudes or Practices. Because of the Likert scale it would be different for Attitudes. You do not describe how “Not sure’s” are handled. You should clearly state for each section the maximum number of points awarded.

• Methods: We need a bit information, based on the previous question on the correlation analysis. Were all percentages or was the correlation analysis done with absolute scores. I would like to see a regression of these data. It is difficult to believe the negative correlation of attitudes with practices, we need to understand that. There is often a gap between knowledge and practice, but I think you might consider a knowledge-Attitude combined score.

• Methods: Need to mention you tried a multi-variate model. My strong suggestion would be to ask for some statistical help on this. Additionally, you should look into logistic regression for your univariate analysis as well. Chi squares to test for independence of two variables but not an association.

• Results: Is a bed net viewed as a preventative measure for dengue? Aedes aegypti is a daybiter so bed nets really are not protective.

• Results: Age and education level look confounded here. You could try a stratified analysis looking to find if you where you look at young people with more education, older people with less education, young people with less education and older people with more education.

• Results: You can’t do anything about it now, but I don’t like your questions associated with breeding sites (larval habitats). No questions about mosquito larvae and if people get they become adult mosquitoes. Also don’t like reference to “broken” containers. Aedes is very happy in intact containers.

• Results: LINES 195-199. WHAT IS A POOR ATTITUDE ON DENGUE AND HOW WAS THAT CALCULATED? IF you add strongly agree with agree, seems like most respondents have an appropriate attitude about dengue, although some of these questions are more about knowledge.

CRITICAL ISSUES.

• How were attitude and practice scores calculated.

• Correlation analysis seems wrong – I can not access the data set which should be saved in csv or txt format.

We cannot make any decision about publication until we have seen the revised manuscript and your response to the reviewers' comments. Your revised manuscript is also likely to be sent to reviewers for further evaluation.

Sincerely,

Amy C. Morrison, PhD

Section Editor

Amy Morrison

Section Editor

Section Editor Comments

I have taken over for Dr. Lenhart and taken the opportunity to provide another review of the revised manuscript. Below I try to provide a roadmap that will get this manuscript accepted for publication as all reviewer’s saw relevance and value in the publication. I have also taken the opportunity to edit the manuscript using the word version provided by the authors. All my suggestions are optional, and my assumption is that most grammatical errors were removed but some additional editing will improve understanding and the flow of the manuscript. Additionally, I include some comments to the review responses, and argue for inclusion of more of those responses in the body of the manuscript.

Previous Review Responses:

Reviewer #1 had a few questions associated with sample size and study design. (Sample size calculation: How did the authors confirm equal participation from the provinces or consider other factors for the selection of samples to generalize the results?, Study population age group: is it 18-22 or up to 25?) and Reviewer #2 asked how many students were approached.

Editor: To better address these concerns more details are needed in the study design and setting section. Principally, how did you approach students and who did you choose? I’m guessing you were able to email students inviting them to go to a link taking you to the online form. Alternatively, some kind of advertisement asking students to participate was distributed. It looks like students were recruited from the top four universities in Nepal. It would be easier to justify your approach, by simply stating that our aim was to understand Knowledge, Attitudes, and Practices of non-health oriented students and we selected these Universities because the had students from the widest cross section of provinces across Nepal and had access to the internet, etc. (not trying to put words in your mouth, but give us the reason you selected this population). This may represent the best and brightest and that would be something you would point out but seems like a reasonable starting point. Getting a perfectly representative sample is always very difficult so don’t claim that was your aim. 

• You need to better describe how you made contact with students and incorporate your response to Reviewer #2 about the response rate to the questionnaire.

• Report the range of age, from your response one still asks if the range is from 19-22 or 19-25. For example, in Table 1 instead of Less than 22 and 22 or above include the ranges, i.e. 19-21 versus 22-25.

• For provinces, perhaps you can included Bagmati and Lumbini (assuming they are closest to the named universities) and Other (you can list the as a footnote).

• When updating this sections, please give the reasons why you selected these 4 universities and what segment of the Nepalese population they are likely to represent.

• State in the methods that the form was administered in English.

• Add your response to Major comment from Reviewer #1. That is add a section to the methods on validation and pretesting of the study instrument.

• Include the definition of non-health students (also in the response to Major comments Reviewer #1) in this section.

• Ethical clearance: I am not as concerned about this, but all individuals who worked with identifiable information (PII) should have approval from each person’s institution or have those institutions defer to Tribhuvan University. Usually this is done through what is called a relying agreement or a letter indicating this was okay with each of the other institutions.

• In the Data management and analysis section, you should state that you carried out a multivariate analysis, but no significant differences were found and will not be discussed more, however this suggests that some of your univariate findings are due to confounding and should be interpreted with caution.

• In addition, to the inclusion of the questionnaire, you should indicate what you scored as a “correct” answer. In regard to this in Table 2, perhaps you could highlight statements that are incorrect in another color to distinguish with statements that are true. I will put the incorrect statements in red.

Other comments and observations:

• “youngest” emerging infection. Note sure what you mean by that. Do you a “recent”?

• Abstract – see some of my edits in yellow highlight or by comment. I have some concerns about Lines 40-42 and Lines 43-49.

• Introduction: see edits in yellow highlight.

• Methods: While you clearly describe how knowledge was scored (80%), it is not clear how you scored Attitudes or Practices. Because of the Likert scale it would be different for Attitudes. You do not describe how “Not sure’s” are handled. You should clearly state for each section the maximum number of points awarded.

• Methods: We need a bit information, based on the previous question on the correlation analysis. Were all percentages or was the correlation analysis done with absolute scores. I would like to see a regression of these data. It is difficult to believe the negative correlation of attitudes with practices, we need to understand that. There is often a gap between knowledge and practice, but I think you might consider a knowledge-Attitude combined score.

• Methods: Need to mention you tried a multi-variate model. My strong suggestion would be to ask for some statistical help on this. Additionally, you should look into logistic regression for your univariate analysis as well. Chi squares to test for independence of two variables but not an association.

• Results: Is a bed net viewed as a preventative measure for dengue? Aedes aegypti is a daybiter so bed nets really are not protective.

• Results: Age and education level look confounded here. You could try a stratified analysis looking to find if you where you look at young people with more education, older people with less education, young people with less education and older people with more education.

• Results: You can’t do anything about it now, but I don’t like your questions associated with breeding sites (larval habitats). No questions about mosquito larvae and if people get they become adult mosquitoes. Also don’t like reference to “broken” containers. Aedes is very happy in intact containers.

• Results: LINES 195-199. WHAT IS A POOR ATTITUDE ON DENGUE AND HOW WAS THAT CALCULATED? IF you add strongly agree with agree, seems like most respondents have an appropriate attitude about dengue, although some of these questions are more about knowledge.

CRITICAL ISSUES.

• How were attitude and practice scores calculated.

• Correlation analysis seems wrong – I can not access the data set which should be saved in csv or txt format.

Reviewer's Responses to Questions

**Key Review Criteria Required for Acceptance?**

**Methods**

-Are the objectives of the study clearly articulated with a clear testable hypothesis stated?

-Is the study design appropriate to address the stated objectives?

-Is the population clearly described and appropriate for 

---

## [Editor Report · Decision Letter 2]

11 Apr 2024

Dear Mr. Rajbanshi,

Thank you very much for submitting your manuscript "Knowledge, attitude, and practice regarding dengue among non-health undergraduate students of Nepal" for consideration at PLOS Neglected Tropical Diseases. As with all papers reviewed by the journal, your manuscript was reviewed by members of the editorial board and by several independent reviewers. In light of the reviews (below this email), we would like to invite the resubmission of a significantly-revised version that takes into account the reviewers' comments. 

Please see my responses in red to your response document.

Overall the manuscript is much improved, but I must insist that you redo your statistical analysis. At a minimum you need to present Univariate logistic regression instead of Chisquare analysis. Conclusions will probably be very similar. You need to provide some evidence that you tried to build an appropriate multivariate model, using some model building strategy (Forward selection / backward selection or some alternative). At some point you should have a model if only a few variable stay in.

You need to explore the relationships between Age and Education, you have the data to do so, as suggest in my review graph it and look to see if anything jumps out, but it is in no way beyond the scope of your analysis.

See my suggestion on the correlation analysis. I can live with it but you should not talk about statistically insignificant results like they are significant. No significant correlation and stop there.

Please address my remaining queries in red.

I've also included a word version with edits. There was some very wierd formating issues so I just had to rescue the text. Just compare this version to your next version and I think most of my changes are appropriate.

All the best,

Amy

We cannot make any decision about publication until we have seen the revised manuscript and your response to the reviewers' comments. Your revised manuscript is also likely to be sent to reviewers for further evaluation.

Sincerely,

Amy C. Morrison, PhD

Section Editor

Amy Morrison

Section Editor

Please see my responses in red to your response document.

Overall the manuscript is much improved, but I must insist that you redo your statistical analysis. At a minimum you need to present Univariate logistic regression instead of Chisquare analysis. Conclusions will probably be very similar. You need to provide some evidence that you tried to build an appropriate multivariate model, using some model building strategy (Forward selection / backward selection or some alternative). At some point you should have a model if only a few variable stay in.

You need to explore the relationships between Age and Education, you have the data to do so, as suggest in my review graph it and look to see if anything jumps out, but it is in no way beyond the scope of your analysis.

See my suggestion on the correlation analysis. I can live with it but you should not talk about statistically insignificant results like they are significant. No significant correlation and stop there.

Please address my remaining queries in red.

I've also included a word version with edits. There was some very wierd formating issues so I just had to rescue the text. Just compare this version to your next version and I think most of my changes are appropriate.

All the best,

Amy
---

## [Editor Report · Decision Letter 3]

16 May 2024

Dear Mr. Rajbanshi,

We are pleased to inform you that your manuscript 'Knowledge, attitude, and practice regarding dengue among non-health undergraduate students of Nepal' has been provisionally accepted for publication in PLOS Neglected Tropical Diseases.

Best regards,

Amy C. Morrison, PhD

Section Editor

Amy Morrison

Section Editor

Congratulations.

I appreciate you taking most of my suggestions. There is still a few places with awkward grammar, also you do not refer to your supplementary material in the text and should. Because I know the people in production will ask for this I will pass them on to you. I hope you agree that it was worth doing the multivariate analysis, providing a lot of clarity. Thanks for putting up with my meddling but the "knowledge gap" is so great throughout the world despite many efforts and that is only step 1 as you show no association between knowledge/attitudes and knowledge/practices. I wish you luck using this information to better inform your national dengue control program.

All the best,

Amy

---

## [Editor Report · Acceptance letter]

22 May 2024

Dear Mr. Rajbanshi,

We are delighted to inform you that your manuscript, "Knowledge, attitude, and practice regarding dengue among non-health undergraduate students of Nepal," has been formally accepted for publication in PLOS Neglected Tropical Diseases.

Best regards,

Shaden Kamhawi

co-Editor-in-Chief

Paul Brindley

co-Editor-in-Chief
